# Formulation, Optimization, and Evaluation of *Moringa oleifera* Leaf Polyphenol-Loaded Phytosome Delivery System against Breast Cancer Cell Lines

**DOI:** 10.3390/molecules27144430

**Published:** 2022-07-11

**Authors:** Jecinta Wanjiru, Jeremiah Gathirwa, Elingarami Sauli, Hulda Shaid Swai

**Affiliations:** 1Department of Global Health and Biomedical Sciences, School of Life Science and Bioengineering, Nelson Mandela African Institution of Science and Technology, P.O. Box 447, Arusha 23100, Tanzania; elingarami.nkya@nm-aist.ac.tz (E.S.); hulda.swai@nm-aist.ac.tz (H.S.S.); 2Centre for Traditional Medicine and Drug Research, Kenya Medical Research Institute, P.O. Box 54840, Nairobi 00200, Kenya; jgathirwa@kemri.org

**Keywords:** polyphenols, natural nanoparticles, *Moringa oleifera*, antiproliferation, breast cancer

## Abstract

*Moringa oleifera* leaf polyphenols (Mopp) were encapsulated with phytosomes to enhance their efficacy on 4T1 cancer cell lines. The Mopp were extracted via microwave-assisted extraction. *Moringa oleifera* polyphenol-loaded phytosomes (MoP) were prepared with the nanoprecipitation method and characterized using the dynamic light scattering and dialysis membrane techniques. The in vitro cytotoxic and antiproliferative activity were investigated with the (3-[4,5-dimethylthiazol-2-yl]-2,5-diphenyltetrazole) MTT assay. Acute toxicity was assessed using *Swiss albino* mice. An MoP particle size of 296 ± 0.29 nm, −40.1 ± 1.19 mV zeta potential, and polydispersity index of 0.106 ± 0.002 were obtained. The total phenolic content was 50.81 ± 0.02 mg GAE/g, while encapsulation efficiency was 90.32 ± 0.11%. The drug release profiles demonstrated biphasic and prolonged subsequent sustained release. In vitro assays indicated MoP had a low cytotoxicity effect of 98.84 ± 0.53 μg/mL, doxorubicin was 68.35 ± 3.508, and Mopp was 212.9 ± 1.30 μg/mL. Moreover, MoP exhibited the highest antiproliferative effect on 4T1 cancer cells with an inhibitory concentration of 7.73 ± 2.87 μg/mL and selectivity index > 3. The results indicated a significant difference (*p* ≤ 0.001) in MoP when compared to Mopp and doxorubicin. The in vivo investigation showed the safety of MoP at a dose below 2000 mg/kg. The present findings suggest that MoP may serve as an effective and promising formulation for breast cancer drug delivery and therapy.

## 1. Introduction

Breast cancer is a major global health problem and a common cause of cancer deaths in women [1]. According to Lei et al. [2], an estimated 2.3 million breast cancer cases and 685,000 deaths in 2020 were reported worldwide. Developing countries reported an estimated 74,072 deaths and 168,690 cases in 2018 [3]. It is one of the most common types of women’s cancer in Sub-Saharan Africa with a projection of 416,000 deaths expected between 2020 and 2029 [4]. In Tanzania, case data demonstrated an upward growth trajectory from 2732 in 2012 to 3037 new cases in 2018 [2] which are anticipated to increase by 82% by 2030 [5]. Without interventions, the burden of breast cancer is bound to rise, exerting tremendous financial and emotional strains on individuals. Adjuvant treatments of breast cancer include chemotherapy, immunotherapy, and radiation therapy. However, they confer adverse side effects, some cancer cells can gain resistance, and they are relatively costly, especially in developing countries [6]. These drawbacks have necessitated the search for alternative efficacious and affordable cancer drugs. In this context, medicinal plants are an important alternative approach since they pose fewer side effects in appropriate doses [7,8]. Therefore, plant-derived agents, for example Docetaxel from *Taxus baccata*, are used in some cancer treatments [9].

*Moringa oleifera* Lam. (Mo) leaf is a commonly used herbal medicine for cancer treatment in African communities. Several in vitro studies have demonstrated its efficacy against breast cancer [10,11,12,13]. The efficacy is attributable to polyphenols such as flavonoids, tannins, and terpenoids. Although polyphenols have antineoplastic effects, they are characterized by poor plasma membrane permeability and lipid immiscibility due to their hydrophilic nature, which limits their absorption into the target cells and also lowers their bioavailability and therapeutic efficacy [10,14,15]. *Moringa oleifera* polyphenols (Mopp) are also degraded by digestive enzymes and gastric fluids when administered orally [15]. An example of a novel drug delivery system using different nanocarriers in the nanomedicine field includes silver conjugated with Mopp to solve the bioavailability challenges, but silver can impose toxicity [16,17]. Therefore, there is a need to search for novel drug delivery systems that are safer and more bioavailable, hence the introduction of nanomedicine in this study.

Nanomedicine is a study area that involves comprehensive monitoring, repair, control, defense, construction, and the improvement of the biological system at a molecular level by using engineered nanostructure and nanodevices to achieve health benefits. Cancer nanomedicine development has focused on improving the efficacy of cytotoxic agents and improving their delivery into tumor cells. Nanomedicine has been tailored to initiating tumor cell death using hybrid nanostructures, such as phytosomes, which act as a drug delivery vehicles for bioactive compounds [18]. Phytosomes form part of nanomedicine that significantly impacts the delivery of anticancer agents.

Phytosomes (ps) comprise a new drug delivery system with an enhanced controlled drug release profile. They are formed of phospholipids, a lipid, and the cell membrane’s major component. Phospholipids are amphipathic molecules with two neutrally charged tails and a positively charged head that renders them miscible in lipid and water conditions. Part of the phosphate group, the oxygen atom, tends to lose or gain electrons [19], which facilitates the modification of polyphenols, generating an amphiphilic complex that expediates their movement across the cell membrane barrier. Therefore, ps can be attractive delivery vehicles to enhance the absorption of polyphenols across biological lipid barriers, thus increasing their efficacy and improving their therapeutic effects. The advantages of phytosomes as a traditional drug delivery system include controlled drug release, low toxicity, biodegradability, biocompatibility, drug distribution, and increased bioavailability of polyphenols. Despite the promising potential for phytosomes, there are few studies that have investigated anticancer-associated phytosomes as a drug delivery system in cancer therapy. Thus, a limited number of products such as SiliphosR, silybin phytosomes, and MerivaR, a curcumin phytosome, are currently in the market. Phytosomes, therefore, being nanocarriers, attract great interest owing to their minimization of the side effects of polyphenols and improvement of their bioavailability.

Previous studies have reported improved anticancer efficacy of nano-phytosomes compared to free polyphenols on breast cancer cell lines [20,21]. Therefore, this study focused on exploring an effective phytosome delivery system that overcomes the bioavailability challenge, limits toxicity, and enhances the antitumor effect in clinical applications. Although *Moringa oleifera* crude extracts have been used in the management of cancer, there are currently no reported studies on the antiproliferative activity of a *Moringa oleifera* phytosomes (MoP) complex on 4T1 cancer cell lines. Thus, this study explored the antiproliferative efficacy of formulated MoP complex on breast cancer cell lines and their in vivo cytotoxic activity using female Swiss albino mice.

## 2. Results and Discussion

### 2.1. Determination of Total Phenolic Content by Use of Standard Curve and the Percentage Encapsulation Efficiency

The ultimate goal of this study was to develop a natural lipid nanoparticle with anticancer properties. Despite having potent antioxidant, liver protective, anti-inflammatory, and anticancer properties, polyphenols are limited in their efficacy due to their poor bioavailability, and this results in a poor clinical trial outcome. This study aimed at preparing Mo polyphenol-loaded phytosomes to overcome this problem without compromising safety. Phytosome technology has the tendency to form intermolecular bonding between phosphotidylcholine and polyphenols. A significant number of polyphenols were successfully extracted and then tested for their TPC.

The plot had a slope (m) = 1.5305 and an intercept of 0.0181. The standard curve equation was y = 1.5305x + 0.0181, R^2^ = 0.998, as seen in Figure 1. The correlation coefficient of (R^2^) =0.9998 was an indication of a good linearity concentration as shown in Figure 1. The gallic acid conformed to Beer’s law with a regression coefficient (R^2^) = 0.9998 since it met the acceptance linearity criteria for a value not less than 0.990 [22].

The concentration ranges used were 1 (100%), 0.5 (50%), 0.25 (25%), 0.125 (12.5%), 0.0625 (6.25%), and 0.03125 (3.125%) µg/mL as shown in Figure 1. The TPC values were calculated based on the above linear equation according to the calibration curve of gallic acid; y = 1.5305x + 0.0181, R^2^ = 0.9998, where X is the amount of gallic acid in µg and y is the absorbance.

The TPC ranged from 45.89 ± 0.27 to 50.81 ± 0.02 mg GAE/g as shown in Table 1 and Figure 2. The phenolic content in the Mo phytosomes was lower than in the Mopp, indicating that not all the polyphenols were encapsulated in the phytosome complexes during optimization and preparation, but a significant quantity was incorporated in the phytosome complex. Typical phenolics have been shown to possess antioxidant, antiproliferation, and anticancer activity alongside being characterized as phenolic acids and flavonoids. In this study, MoP drug delivery system complex was successfully developed. Phytosomes are used to deliver polyphenols to the target. This improves the absorption, bioavailability, control of drug release, and drug distribution, thereby enhancing the passive targeting of cancerous cells, retention of polyphenols, permeability effect, and efficacy, producing better results than the conventional herbal extracts effect.

The TPC of Mopp at a concentration of 1 mg/mL was 50.81 ± 0.02 mg GAE/g, while MoP was 45.89 ± 0.27 mg GAE/g of Mo dry sample as shown in Figure 2 and Table 1. This is an indication of high TPC in the extracted Mopp. This assay was important because high total phenolic content has been shown to inhibit initiation and progression of cancer cells by modulating genes, regulating key processes such as growth and development of tumors, oncogenic, angiogenesis, and metastasis [23]. The perfect number of Mo polyphenols on the phytosome nanoparticles with minimal loss throughout the preparation was reflected by the high % EE as shown in Table 1. Percentage encapsulation efficiency *is* one of important parameters that is used to evaluate the success of the drug delivery system.

The percentage entrapment efficiency of Mopp was verified by the Folin–Ciocalteu spectrophotometric method and validated as shown in Table 1. Previous studies have reported that the formulation’s % EE depends on drug solubility and bonding interaction leading to matrix formation. Therefore, the present data support results from the earlier studies by Pal et al. [23], in which a higher % EE for polyphenols is demonstrated.

### 2.2. Particle Size, Polydispersity Index, and Zeta Potential of Formulated Moringa oleifera Phytosomes

Dynamic light scattering (DLS) is a technique that is based on Brownian motion of the dispersed particles. When these particles are dispersed in liquid they move randomly in all directions and constantly collide with the solvent molecules. These collisions then cause energy transfer, which induces the particle movement. The transferred energy is more or less constant and usually has a greater effect on smaller particles. This results to smaller particles moving at greater speeds than the larger particles. DLS thus allows the measurement of the hydrodynamic diameter of a particle size in a solution. The data shown in Figure 3 and Figure 4 were the final results of five measurements of the hydrodynamic diameter that are presented as mean ± standard deviation in nanometers. The average MoP formulated size was found to be in the range of 137.6 ± 1.47 to 296 ± 0.29 nm, and this agrees with a range of microsphere < 500 nm as seen in Figure 3. The average particle size distribution (PDI) showed that the MoP particles were homogeneously distributed, supported by the PDI value ranging from 0.106 ± 0.002 to 0.204 ± 0.011 as seen in Figure 3. This agrees with Piazzini et al. [24] that the best homogeneity is <0.5.

The zeta potential is an essential parameter that measures colloidal dispersion such as phytosome stability. The zeta potential value obtained for MoP was −40.1 ± 1.19 mV. A higher electrostatic repulsion between the particles indicates higher stability. According to the present findings shown in Figure 4, Zeta potential was higher than −40.1 mV, implying that the prepared MoP had good physical stability.

### 2.3. Fourier Transform Infrared Spectroscopy

The FTIR spectra confirmed the formation of MoP as shown in Table 2 and Figure 5a–c.

The FTIR analysis was used to study the possible interactions between polyphenols of Mo and phospholipids. Infrared (IR) spectra were recorded in the range of 4000 to 400 cm^−1^. The obtained IR spectra were interpreted for presence of functional groups from polyphenols, phosphotidylcholine, and phytosomes at their respective wavenumbers (cm^−1^) as indicated in Figure 5 and Table 2. The spectra showed a broad peak at 3271 cm^−1^, representing the aliphatic alcoholic (−OH) group usually present in polyphenolic groups. Phospholipids shielded the peaks at 1613 cm^−1^ (C=C), indicating the presence of unsaturated compounds usually found in fatty acids in the phytosomes, confirming successful synthesis of phytosomes. The spectra also showed fatty acidsꞌ long-chain bands of the phospholipid molecule at 2958, 2923, and 2855 cm^−1^ also indicating phytosome formation as shown in Figure 5. The band at 1257 cm^−1^ indicated the association of C-N stretching vibration usually found in the choline moiety in the phytosomes. The spectral band at 1023 cm^−1^ is attributed to symmetric stretching of the C-O-C vibration mode present in the phospholipids and polyphenols, and this indicates the successful formation of the Mo phytosome complex. The absorption at 1613 cm^−1^ in the complex spectrum indicates the formation of hydrogen bonds and the existence of electrostatic interactions between polyphenols and phospholipids as shown in Figure 5a, all of which confirm the successful formation of the phytosome.

### 2.4. In Vitro Drug Release

The in vitro drug release of optimized MoP complex and Mopp is shown in Figure 6, while the drug release for different kinetic models with their correlation coefficients is shown in Table 3 and Table 4.

To analyze the mechanism of MoP and how the polyphenols are released from the drug delivery system, the polyphenols for MoP complex at different times were compared. The in vitro polyphenol drug release profile studies for MoP and Mopp indicated a sustained and controlled release. A fast polyphenol release (43.43%) was observed at the end of the 8th h, followed by a sustained drug release (53.49%) over the remaining 24, 48, and 72 h for the MoP as observed in Figure 6. Although a burst release profile was observed in both MoP and Mopp profiles, the release was higher in the Mo phytosome formulation as shown in Figure 6. The initial drug release burst was due to the drug being entrapped near the surface, while the sustained release was attributed to its release diffusion from the phytosomes. The in vitro release study data were fitted to different kinetic models, and the best fit was achieved by the highest correlation coefficient. The Korsmeyer–Peppas model release kinetics with linear regression R^2^ = 0.9306 was the best fit for this study as indicated in Table 3 and Table 4. The drug release according to the Korsmeyer equation shows that the diffusion and erosion could be the drug release mechanism, agreeing with the results of Parashar et al. [25].

In our study, ‘in vitro release testing” was used as a measure of drug dissolution as documented previously by Shen and Burgess [26]. This is considered an important tool for quality control purposes as well as for prediction of the in vivo performance of drug delivery involving nanocarrier systems and has become an essential quality control test of drug development since it was officially adopted in the United State Pharmacopeia (USP) in 1970 [27,28]. There is no standard pharmacopeial/regulatory in vitro dissolution/release test currently available for nanoparticulate systems [26]. However, extensive efforts have been made to develop suitable in vitro dissolution/release testing methods for nanoparticulate delivery systems. Current methods are broadly divided into three categories, namely membrane diffusion methods (such as dialysis methods), sample and separation methods, and continuous flow methods.

Membrane diffusion methods (dialysis methods) are the most widely investigated for the in vitro dissolution release testing of nanoparticulate systems. In these methods, the nanoparticulate systems are separated from the release medium through dialysis membranes that are permeable to the free drug but impermeable to the nanoparticles. Dialysis methods have been widely used to investigate in vitro drug dissolution/release profiles of liposomes [29,30], emulsions [31], polymeric nanoparticles [32,33], as well as lipid nanocarriers [34].

### 2.5. In Vitro Bioaccessibility

As shown in Figure 7, there was a significant difference in percentage bioaccessibility of polyphenols in the GIT when TPC of Mopp was compared to MoP (*p*-value 0.001) after in vitro digestion. Therefore, these results proved the successful effect of phytosomes as a carrier (Figure 7).

The free Mopp bioaccessibility mean percentage was 26.95 ± 0.02, showing extensive degradation, whereas the MoP bioaccessibility mean percent was 66.98 ± 0.01%; indicating a better stability than the Mopp. The low bioaccessibility of Mopp was due to compromised conditions (digestive enzymes and the acidic environment) of the gastrointestinal tract. These findings are in line with previous reports where low bioaccessibility of free phenolic compounds was seen after in vitro digestion as compared to encapsulated polyphenols [35,36]. This might be attributed to low absorption due to larger molecular weight and thus limited bioavailability.

### 2.6. Physical Storage Stability Test

Table 5 shows the in vitro stability test for MoP carried out at room temperature for 25 days. The MoP average size, zeta potential, and particle distribution, PDI was used to evaluate the physical change of the phytosome complex.

The MoP formulations characterization parameters by DLS technique after day 1 to 25 days of storage at 25 °C showed an average zeta size range of 220.3 ± 0.12 to 239.6 ± 2.46 nm. The Zeta potential ranged between −38.3 ± 1.14 to −42.8 ± 2.53 mV while the PDI ranged between 0.11 ± 0.022 and 0.19 ± 0.065 as shown in Table 5. There was no significant change in MoP particle average size, zeta potential and polydispersity index for the 14 days (*p*-value > 0.05). The storage stability of Mo phytosomes results was presented at 25 °C. There was a small variation in average size, PDI and Zeta potential over the 25 days (*p* ˃ 0.05), which shows that the sample remained stable when exposed at room temperature. This shows that the formulated phytosomal complex retained its charge; and the tendency for aggregation at 25 °C.

### 2.7. Phytosomes, Polyphenols, and Doxorubicin Effects on Vero E6 (Normal) Cell Lines

The criteria usually used for in vitro cytotoxicity after 72 h of exposure time for the US National Cancer Institute indicate that an IC_50_ < 4 µg/mL for pure compounds and IC_50_ < 20 µg/mL for crude extracts are considered highly antiproliferative. Additionally, an IC_50_ value less than 30 µg/mL is considered to be antiproliferative, while an IC_50_ value of 30–100 µg/mL is considered moderately antiproliferative, and above 100 µg/mL is indicated as inactive [37].

The Mopp and MoP had CC_50_ > 90 μg/mL in the cytotoxicity studies except for doxorubicin. The CC_50_ value of the Vero cell line ranged between 68.35 ± 3.51 and 212.9 ± 1.30 μg/mL (Figure 7). The inhibitory concentration for doxorubicin, MoP complex, and free Mopp was 3.04 ± 0.27, 7.73 ± 2.87, and 39.84 ± 0.10. The selectivity index for doxorubicin, MoP and Mopp was 2.07 ± 1.30, 12.79 ± 0.50, and 5.34 ± 0.13 μg/mL, respectively, as seen in Figure 8.

The results in Figure 8 showed that the MoP was nontoxic to normal cells and highly selective to breast cancer cells compared to free polyphenols. Doxorubicin was toxic to 4T1 cell lines as the selectivity index was ≤ 3, and this is because according to the National cancer institute, selectivity index below 3 indicates higher cellular toxicity. Generally, the MoP inhibited the proliferation of 4T1 cancer cells. According to the findings in Figure 9, a low concentration of MoP at 0.14 μg/mL did not affect cell growth substantially, whereas the treatment with 0.41, 1.23, 3.70, 11.11, 33.33, and 100 μg/mL of Mo phytosomes inhibited the growth of 4T1 cells in a dose-dependent manner. There was a significant difference between the cytotoxicity of Mopp and MoP (*p* ≤ 0.05), and their activity was not comparable to that of the positive control.

In vitro antiproliferative activity of Mo leaf crude extract has been considered in previous studies [10,11,12,13] and showed an inhibitory concentration above 100 μg/mL. In this study, the inhibitory concentration of Mo polyphenols was 39.94 ± 0.10 μg/mL, while that of MoP complex was 7.73 ± 2.87 μg/mL, which is an indication that the Mopp had an improved efficacy compared to reported earlier Mo crude extract. These results indicated that MoP extracts from *Moringa oleifera* induced apoptosis of 4T1 cancer cells and can be considered as active. This because according to suggestions from the National Cancer Institute (NCI), an IC_50_ equal or lower than 20 μg/mL can be suitable as a benchmark for screening cancer drugs that are from herbs and plants. A study by Xu et al. [38] reported improved inhibitory effects of polyphenols on 4T1 cancer cells. Studies by Liu et al. [39,40,41] have reported improved bioavailability of bioactive compounds, thus improving their potential inhibiting effect on breast cancer cells. The present study findings agree with the abovementioned reported studies, and those of other researchers such as Moeini [21] where phytosomes and other lipid phospholipid nanocarriers improved bioavailability of bioactive compounds, thus improving their efficacy in inhibiting breast cancer cells growth. This study on MoP formulation and antiproliferative effects on 4T1 is being reported for the first time.

### 2.8. In Vivo Toxicity Studies

The oral administration of Mopp and the MoP at doses of 50, 300, and 2000 mg/kg doses resulted in the absence of signs of acute toxicity. The weight range was as shown in Table 6.

The oral administration of free Mopp and Mo phytosomes at a dose from 50 up to 2000 mg/kg resulted in no clinical signs of acute toxicity, nor did they influence the body weight of the Swiss albino mice during a short period of 5 h or in a prolonged period of 14-day observation as shown in Table 6. There was no mortality in either control or treated groups across the different doses of the drugs. Additionally, there were no abnormalities in the autopsy organs or significant body weight change during the 14 days of study. Although the acute toxicity of MoP complex and Mopp in Swiss albino mice is being reported for the first time, the study agrees with the in vitro safety studies of MoP on normal cells. This suggests the in vivo safety of the novel MoP complex at an oral dosage of up to 2000 mg/kg in *Swiss albino* mice.

## 3. Materials and Methods

### 3.1. Reagents and Chemicals

Dichloromethane (99% purity) and ethanol were procured from Sigma-Aldrich Inc. Soy phosphatidylcholine (lipoid) was obtained from Biotec Lab Ltd. The MTT and Trizol ™ reagent were obtained from Sigma. The 4T1 mammary carcinoma cells and Vero cell lines were ordered from the American Type Culture Collection (ATCC) Manassas, VA, USA). Dulbecco’s modified Eagle’s medium (DMEM) was from Gibco (Life Technologies, Inc., Carlsbad, CA, USA). Fetal bovine serum (FBS) 10%, L-glutamine 1%, and 1% antibiotics were purchased from Gibco.

### 3.2. Ethical Considerations

Before commencement of the study, clearance was sought from the Kenya Medical Research Institute (KEMRI), Scientific and Ethics Review Unit (SERU). All the laboratory procedures and protocols were followed, and no human subjects were used in the study.

### 3.3. Sample Collection and Preparation

*Moringa oleifera* leaves were collected from 20–26 May 2021, at Machame, Moshi, Tanzania, where they were being used by the Traditional Health Practitioners (THPs) and local communities for breast cancer management. The collection was carried out with assistance from a botanist and the area herbalist, and an authentication voucher number JWN/MO/05/2021 was awarded by a professional taxonomist. The samples were taken to the Nelson Mandela African Institution of Science and Technology (NM-AIST), sorted, air-dried for seven days, and then pulverized into fine powder. The materials were then filtered using Whatman’s filter paper (No. 1) and subjected to a microwave-assisted extraction technique.

### 3.4. Microwave-Assisted Extraction

A domestic microwave oven (Akai A24001), with 800 W total capacity operating at 2.54 GHz, was employed for Mopp extraction. Approximately 5.0 g of plant samples were mixed with distilled water (100 mL). The mixtures were irradiated at (750 W, 90 s) as described Sánchez Camargo et al. [42] with some modifications. The resulting mixture was then filtered using Whatman’s No. 1 filter paper, concentrated by a freeze drier, and the extracted yield was determined gravimetrically.

### 3.5. Estimation of Total Phenolic Content

The quantification of total phenolic content (TPC) was assessed following the Folin–Ciocalteu method [43] with slight modifications. Triplicate tests were conducted in the experiment. The results were expressed as mg of gallic acid equivalent (GAE) per 100 g of dry MO sample (mg GAE/100 g). The prepared sample was then placed in an air-tight bottle and transferred to the Centre for Traditional Medicine and Drug Research (CTMDR) at the Kenya Medical Research Institute (KEMRI) laboratories in Kenya for further analysis.

### 3.6. Phytosome Synthesis

A *Moringa oleifera* phytosome formulation was prepared using a thin-layer hydration method [44]. Briefly, lipoid was dissolved in dichloromethane, while Mopp were diluted with 90% ethanol. The mixture was then poured into a round-bottom flask. Sonication followed for 10 min, and a BUCHI Mini spray drier B−290 (Inlet and outlet temperature; 100 °C, pump rate 25; aspirator; 100) was used to flow nitrogen gas which led to solvent evaporation. An even and a thin film layer MoP complex was finally formed.

### 3.7. Standard Curve and Percentage Entrapment Efficiency

The Folin–Ciocalteu method using gallic acid was used to draw a standard curve that was then used to calculate total phenolic content for MoP and free Mopp. The method is based on the transfer of electrons in an alkaline medium from phenolic compounds to phosphomolybedic phosphotungstic acid complexes in Folin to form blue colored complexes (PmoW11O40)-4 as determined by spectrophotometry at 760 nm. The calibration curve was used to measure the linearity of the method.

The percentage entrapment (% EE) of Mopp in the phytosome complex was determined by the ultracentrifugation method as previously described by El-Fattah et al. [45,46,47]. The MoP formulations were briefly centrifuged at 4 °C and 15,000 rpm for 90 min. The supernatant was then separated and analyzed for TPC as previously described using a UV spectrophotometer at 290 nm.

### 3.8. Particle Size Distribution, Zeta Potential, and Polydispersity Index

The particle size distribution, zeta potential, and polydispersity index (PDI) for the MoP complex were assessed using the dynamic light scattering (DLS) technique with a particle size analyzer comprising a Malvern Zetasizer Nano computerized system using the protocol of El-Far et al. [45]. The following parameters were used: temperature of 25 °C, a wavelength of 633 nm, 173° light scattering angle, a 1.33 refractive index of the medium, and 0.8872 cP medium viscosity.

### 3.9. Fourier Transform Infrared Spectroscopy

The Fourier transform infrared (FTIR) (JASCO 4700 ATR-FT/IR) spectral analysis of MoP complex was used to determine the chemical stability and structure of the compound using the protocol previously described by Thiruvengadam and Bansod [48]. Potassium bromide pellets were freshly prepared to avoid any moisture effect, mixed with 0.5 mL of the sample, then placed below the fixed probe of FTIR and scanned over a spectrum of 4400 to 400 cm^−^^1^ wavenumber region.

### 3.10. In Vitro Drug Release of Polyphenol from Moringa oleifera Phytosomes

In vitro drug release of Mo phytosomes was analyzed using a dynamic dialysis method with slight modifications [49]. A dialysis bag at 4000 Da was used with phosphate-buffered saline (PBS) at 0.02 M concentration and 7.4 pH. The dialysis bags allowed the released polyphenols to permeate the release medium. Briefly, an equivalent of 10 mg of Mo phytosomes was dispersed in 2.0 mL of PBS, added into a dialysis bag, and then changed into an Erlenmeyer flask with 100 mL PBS media. The system was then kept on a magnetic stirrer at 37 °C at 100 rpm under controlled conditions. The release media were wholly withdrawn and replaced with fresh PBS solution at designated time intervals from 0 to 8 h, then at 12, 24, and 72 h. The release media (media with the samples) were filtered using 0.45 Millipore filter paper and measured against fresh PBS media as blank at λmax of 425 nm by spectrophotometry. The percentages of polyphenol release were plotted as a function of time according to different kinetics models (zero-order, first-order, Higuchi, and Korsmeyer–Peppas) [50].

### 3.11. In Vitro Bioaccessibility Determination of MoP and Mopp

The MoP and Mopp were evaluated for their bioaccessibility in a simulated gastrointestinal (GIT) model consisting of the mouth, stomach, and intestine according to the method of Grgić et al. [36]. The prepared samples were then exposed to the simulated gastric and small intestine phases.

#### 3.11.1. Simulated Salivary Fluid in Mouth Phase

Simulated salivary fluid phase (SSF) was prepared using 0.328 g/L ammonium nitrate, 1.594 g/L sodium chloride, 0.202 g/L potassium chloride, 0.636 g/L potassium phosphate, 0.198 g/L urea, 0.308 g/L potassium citrate, 0.146 g/L lactic acid sodium salt and 5 g/L porcine gastric mucin type II. An aliquot of 4 mL of each extract was mixed with 4 mL of simulated saliva and the pH of the mixture adjusted to 6.8. The mixture was shaken continuously for 10 min at 100 rev while maintaining temperature at 37 °C.

#### 3.11.2. Simulated Gastric Fluid (SGF)

The SGF was prepared with a slight modification of the methods of Shah et al. [35] and Grgić et al. [36]. Two grams of sodium chloride, 7.0 mL of hydrochloric acid (420 g/L) and pepsin (3.2 g) was dissolved in 1 L of double distilled water and the pH adjusted to 1.2 using 1 MHCl. The sample from the mouth phase mimicking the bolus was mixed with SGF phase at a ratio of 50:50. The pH of the two-phase mixture was adjusted to 2.0 using 1 M NaOH and incubated at 37 °C for 2 h with continuous shaking at a speed of 100 rev/min.

#### 3.11.3. Small Intestinal Phase

About 15 mL of the digested sample from the gastric phase was mixed with 8.25 mL simulated intestinal buffer solution. An aliquot of 1.87 mL fresh bile extract, 30 µL of 0.3 M calcium chloride, and 3.75 mL of pancreatin solution were also added and the volume topped up to 30 mL using deionized water. The temperature was maintained at 37 °C and pH adjusted to 7.0 with 1 M NaOH. Approximately 1.5 mL lipase suspension (at a concentration of 60 mg/mL) was dissolved in phosphate-buffered saline (PBS) and added to the mixture. The mixture was then allowed to shake for 2 h while monitoring the pH (pH was maintained at 7.0) to mimic intestinal digestion process. NaOH (0.25 M) was used to neutralize the fatty acids released from the lipid digestion while maintaining the pH of 7.0. The mixture was then shaken at 100 rpm for 6 h at 37 °C.

#### 3.11.4. Measurement of Bioaccessibility

At the end of in vitro digestion, digested sample was used to measure the percentage bioaccessibility. Triton X-100 (1%) was added to the MoP sample to rupture the lipid membrane, and the mixture was vortexed and later centrifuged at 20,000 rpm for 30 min at 4 °C. The supernatant was collected and filtered. The filtrate was then fractioned and phenolic compounds solubilized. The total phenolic content was quantified and bioaccessibility calculated as follows:
(1)Bioaccessibility (%) = CDigesta/Ci × 100


### 3.12. In Vitro Storage Stability Tests

A thin layer hydration method was used in the MoP formulation. A short-term stability test of MoP complex was then evaluated immediately after preparation, subsequently at regular time intervals, according to Lang et al. [51]. Briefly, 10 mg of MoP complex was stored at 25 °C room temperature for 25 days. As an essential indicator to assess the short-term stability of MoP, the drug-loading residual content was evaluated at predetermined time intervals (0 to 8 and 25 days).

### 3.13. Cell Viability Test

Cell cytotoxicity of MoP complex against 4T1 cancer and Vero (E6) cell lines was assessed by MTT assay according to the protocol of Alhakamy et al. [52]. Briefly, a monolayer in the exponential growth phase was trypsinized, after which trypan blue was used to count the viable cells. Cells (1 × 10^6^ cells/mL) were seeded in a 96-well microtiter plate in minimum essential media with fetal bovine serum (100 μL) and incubated in 5% CO_2_ at 37 °C. After 24 h, the cells were exposed to 20 μL of MoP added in triplicate at the starting concentrations of 100 μL followed by three-fold serial dilutions and incubated for 72 h. Thereafter, 50 μL of MTT dye was added and set at 37 °C for 2 h. After that, approximately 100 μL of dimethyl sulfoxide (DMSO) was added into the solubilized formazan crystals, and absorbance was read at 570 nm using a 96-well microplate reader with a Thermo Fisher MultiscanGo Spectrophotometer model. Doxorubicin was used as a standard drug (positive control). The percentage cytotoxicity was determined using untreated cells as the negative control and expressed in CC_50_ values to infer a concentration that altered 50% of intact cells. In addition, the half-maximal inhibitory concentration (IC_50_) of 4TI by MoP was assessed and calculated using GraphPad Prism 6 Software, USA. The experiment was carried out in triplicate.

### 3.14. In Vivo Acute Toxicity Study Using Swiss albino Mice

Six to eight-week-old female *Swiss albino* mice (18–22 g) were acquired from the KEMRI animal house. They were then housed under standardized conditions before the experiment. A total of 63 mice were weighed, randomly selected, and divided into nine groups. The mice were marked on their tails for easy identification. Additionally, the animals were fed with water and mice food pellets ad libitum. The cages were kept at 25 °C while ensuring lighting regulation. The acute toxicity experiment was carried out according to the Organization for Economic Cooperation and Development (OECD) and slight modifications of the protocol of Laure et al. [53].

Briefly, the mice were fasted for 3 h before dosing and given ad libitum water only. An estimated 0.2 mL of drugs was prepared in PBS at 50, 300, and 2000 mg/mL concentrations based on 1 mL/kg of the mice’s body weights. The mice were observed for general behavior, body weight changes, and mortality for the first 5 h and subsequently every day for 14 days after treatment.

The following parameters were considered: mortality, signs of acute toxicity, and behavioral changes (aggression, paralysis, unusual vocalization, agitation, sedation, tremors, convulsions, ataxia, diarrhea, piloerection, catatonia, unusual locomotion, grooming, fasciculation, sleep, coma, prostration and asphyxia, hypo and hyperactivity, and tremors) [33]. All mice were weighed immediately after treatment on the 1st day then on the 7th and 14th days. At the end of the experiment period, all animals were euthanized, and the spleen, heart, liver, kidneys, lungs, ovaries, intestines, and brains were dissected from each mouse and observed for abnormalities.

### 3.15. Data Management and Statistical Analysis

The raw data were transferred into Microsoft Excel and used in calculating mean absorbance values. Origin 2019b software was used in calculating the total phenolic content. GraphPad software was used to calculate concentrations needed to inhibit 50% cell growth. Quantitative values obtained per treatment were converted to percentage inhibition. All the data are shown as mean ± SEM, the standard error of the mean. ANOVA was performed to compare the models and the insignificant lack of fit for each output factor depended on the *p*-value. The very small probability values of the selected model of each output factor reflected that the model was significant for the data set (*p* < 0.05). Differences in *p*-value < 0.05 were considered statistically significant. Figures, graphs, and tables were used to give a clear presentation of the data obtained from the study.

## 4. Conclusions

In the present study, the MoP complexes were successfully formulated and characterized, and they showed improved bioavailability of Mopp, especially in inhibiting 4T1 cancer growth. This is an indication that the MoP complexes can be a reliable candidate for improved drug dosage in breast cancer therapy. This study concludes that the novel MoP have better physical characteristics and reduced cytotoxicity compared to free Mopp, and they also possess antiproliferation potential on breast cancer cell lines than free MoP polyphenols. The strong antiproliferation activity could be attributed to the high content of polyphenol compounds encapsulated by the present phytosomes that could enhance apoptosis. The MoP phytosomes were also safe in the in vivo study at up to 2000 mg/kg dosage. This dose can be used for further studies in determining the in vivo antiproliferative activity of MoP phytosome complex. The mechanism of action in the in vitro studies on 4T1 is therefore recommended, to identify possible genes involved in inhibiting breast cancer cell growth.

## Figures and Tables

**Figure 1 molecules-27-04430-f001:**
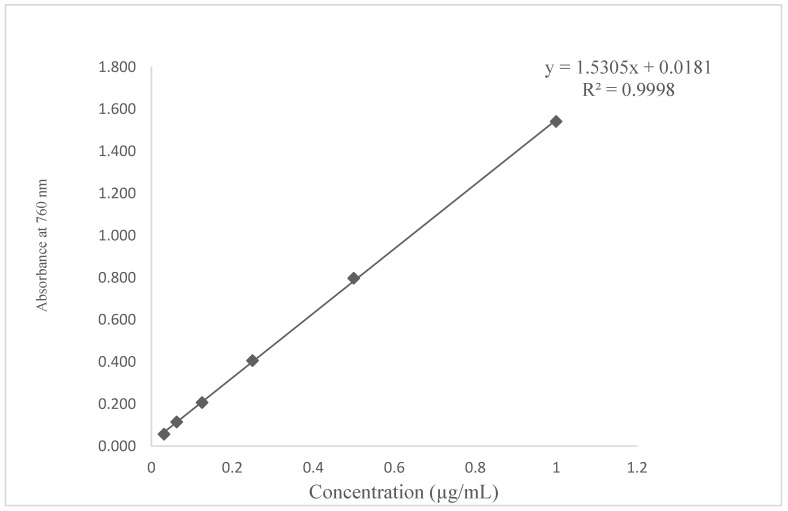
Total phenolic content of standard gallic acid (R^2^ values are a representation of the mean data set of n = 3.

**Figure 2 molecules-27-04430-f002:**
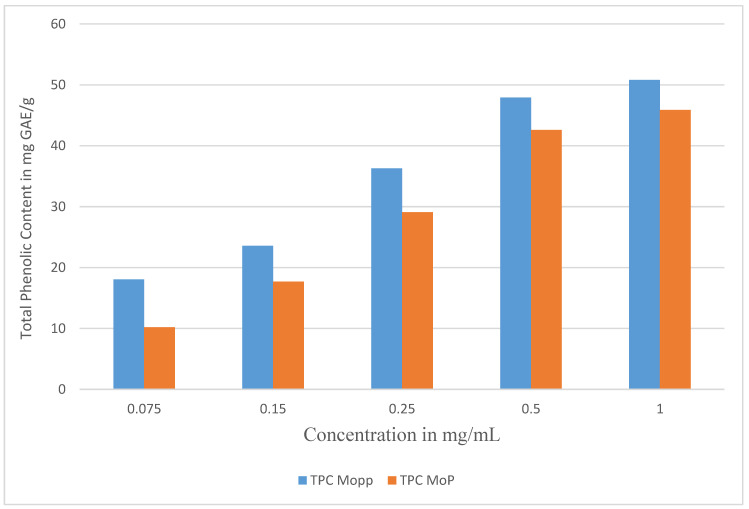
The total phenolic content of Mopp before and after MoP complex formulation.

**Figure 3 molecules-27-04430-f003:**
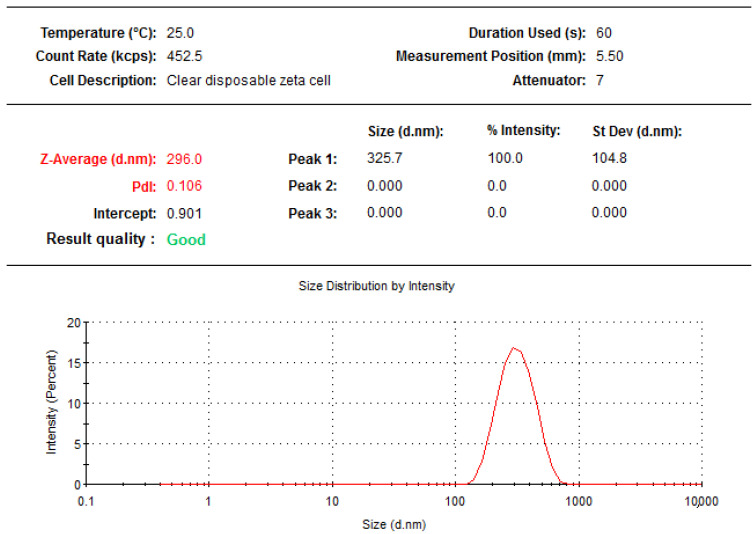
The average particle size and PDI of optimized MoP formulation.

**Figure 4 molecules-27-04430-f004:**
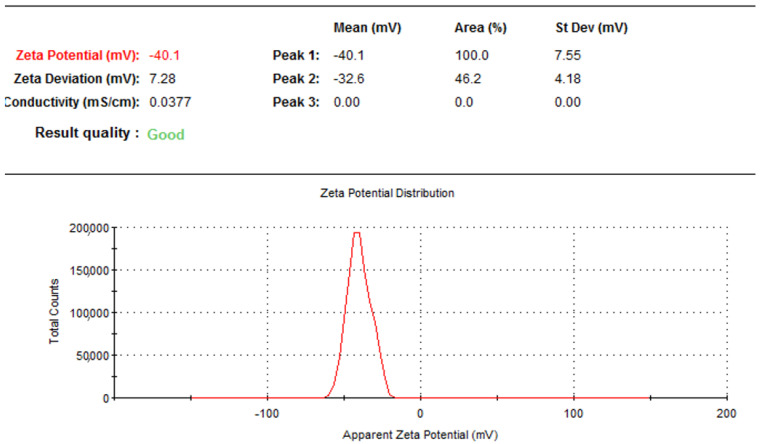
Zeta potential of optimized MoP formulation.

**Figure 5 molecules-27-04430-f005:**
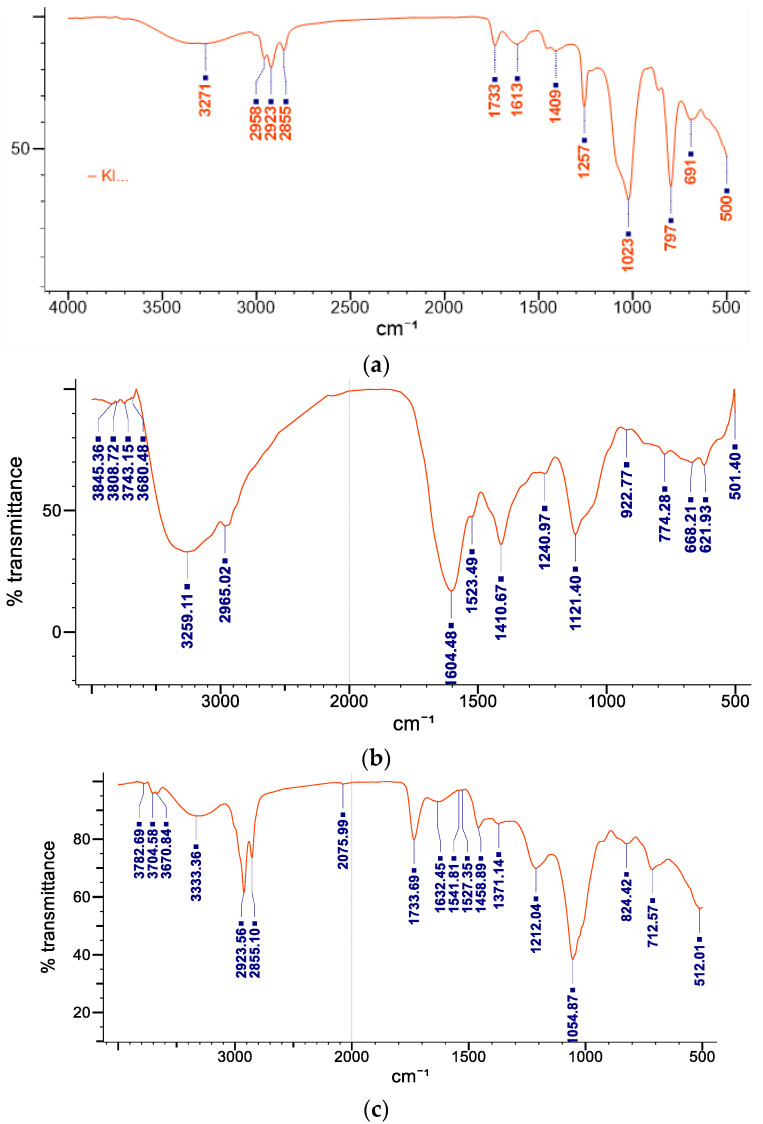
FITR spectra of MoP (**a**), Mopp FTIR spectra (**b**), and phospholipids (**c**).

**Figure 6 molecules-27-04430-f006:**
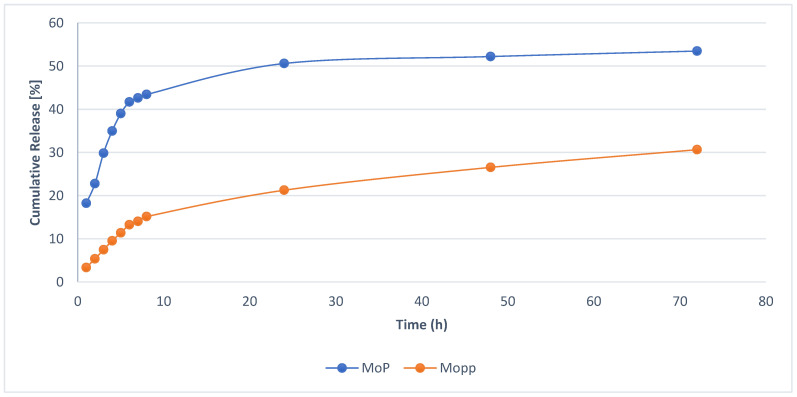
In vitro release profile of optimized MoP compared to free Mopp in phosphate-buffered saline pH 7.4 at 37 ± 0.5 °C (mean ± SD, n = 3).

**Figure 7 molecules-27-04430-f007:**
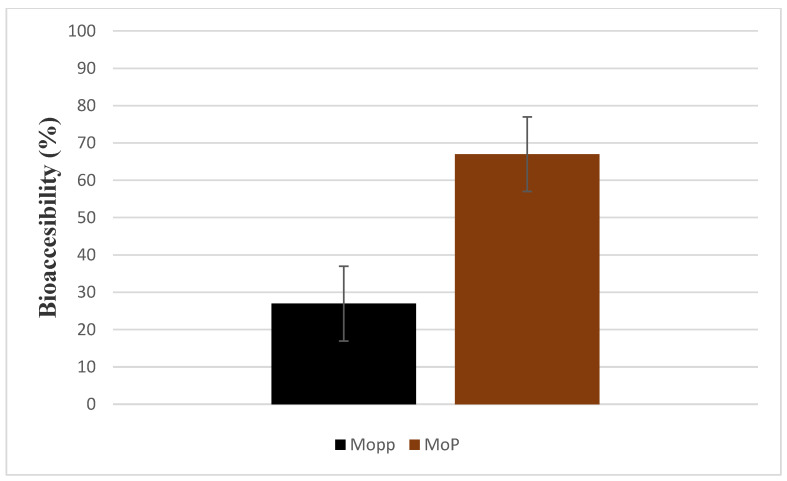
Bioaccessibility comparison of Mopp and MoP after exposure to simulated gastrointestinal conditions.

**Figure 8 molecules-27-04430-f008:**
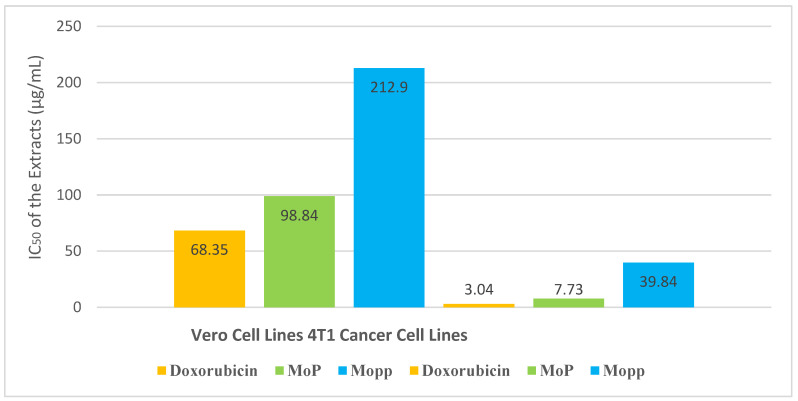
CC_50_ (concentration of the drug that reduces cell viability by 50%) and IC_50_ (concentration that inhibits 50% of breast cancer cells) of doxorubicin (standard drug) and extract (Mopp and MoP) comparisons toward Vero cell lines and 4T1 cell lines.

**Figure 9 molecules-27-04430-f009:**
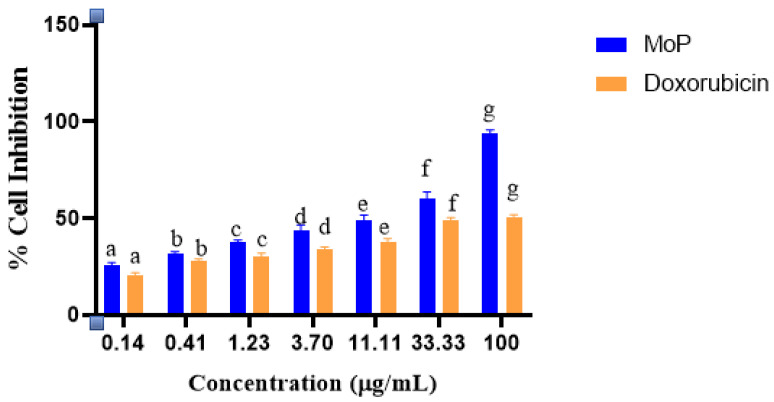
Percentage cell inhibition comparison for MoP and doxorubicin at different dosages. Bars with different letters per concentration are significantly different (*p* < 0.05) according to Student’s t-test analysis.

**Table 1 molecules-27-04430-t001:** Total phenolic content and the encapsulated efficiency.

Concentration mg/mL	TPC Polyphenols in GAE/g	TPC Phytosome GAE/g	% Encapsulation Efficiency
1	50.81 ± 0.02	45.89 ± 0.27	90.32 ± 0.11
0.5	47.93 ± 0.13	42.60 ± 0.33	88.88 ± 0.41
0.25	36.29 ± 0.05	29.11 ± 1.13	80.21 ± 2.26
0.15	23.59 ± 0.17	17.71 ± 0.71	75.07 ± 2.5
0.075	18.07 ± 1.12	10.19 ± 0.13	56.39 ± 1.35

**Table 2 molecules-27-04430-t002:** FTIR and the functional groups present in MoP.

S/No.	Frequency Range (cm^−1^)	Functional Group Identified
1	3271	hydroxyl compound
2	2958, 2923, and 2855	CH and CH2 stretching aliphatic group
3	1733	carbonyl group
4	1613	C=C unsaturated compounds
5	1409	stretching -C=O inorganic carbonate
6	1257	C-N amide 111 band
7	1023	C-O-C group
8	797	C-H
9	691	C-S linkage

**Table 3 molecules-27-04430-t003:** In vitro drug release data for different kinetic models.

Time	Square Root of Time	Log	Cumulative Percentage Drug Release ± SD	Log Cumulative Percentage Drug Release	Cumulative Percent Drug Remaining	Log cumulative Percent Drug Remaining
h		Time				
1	1	0	18.23 ± 0.01	1.26 ± 0.01	81.77 ± 0.41	1.91 ± 0.72
2	1.41	0.3	22.77 ± 0.03	1.36 ± 0.06	77.23 ± 0.91	1.89 ± 0.11
3	1.73	0.48	29.83 ± 0.07	1.47 ± 0.04	70.17 ± 2.36	1.85 ± 0.51
4	2	0.6	32.96 ± 0.18	1.52 ± 0.02	67.04 ± 1.28	1.83 ± 0.02
5	2.24	0.7	40.01 ± 0.15	1.6 ± 0.13	59.99 ± 1.37	1.78 ± 0.31
6	2.45	0.78	41.67 ± 0.90	1.62 ± 0.20	58.31 ± 2.21	1.77 ± 0.95
7	2.65	0.85	42.63 ± 1.07	1.63 ± 0.73	57.37 ± 1.02	1.76 ± 0.91
8	2.83	0.9	43.43 ± 0.47	1.64 ± 1.07	56.57 ± 1.50	1.75 ± 0.07
24	4.9	1.38	50.6 ± 1.25	1.7 ± 0.60	49.4 ± 1.23	1.69 ± 0.09
48	6.93	1.68	52.21 ± 0.64	1.72 ± 0.29	47.79 ± 0.91	1.68 ± 0.32
72	8.49	1.86	53.49 ± 1.02	1.73 ± 0.11	46.51 ± 0.26	1.67 ± 0.02

**Table 4 molecules-27-04430-t004:** Correlation coefficients for different kinetic models.

Formulations	Zero-Order (R^2^)	First-Order (R^2^)	Higuchi (R^2^)	Korsmeyer–Peppas (R^2^)
Phytosome	0.5203	0.59	0.8877	0.9306

**Table 5 molecules-27-04430-t005:** The in vitro storage stability for MoP at 25 °C, room temperature. The data are expressed as mean ± SD.

Number in Days	Average Particle Size in nm	Zeta Potential in mV	Polydispersity Index
1	220.3 ± 0.12	−38.3 ± 1.14	0.11 ± 0.02
5	227.9 ± 1.11	−40.9 ± 3.56	0.13 ± 0.11
10	229.6 ± 0.20	−41.1 ± 2.48	0.14 ± 0.04
15	231.7 ± 1.34	41.4 ± 1.52	0.17 ± 0.16
20	236.5 ± 2.53	−42.7 ± 31	0.19 ± 0.03
25	239.6 ± 2.46	−42.8 ± 2.53	0.19 ± 0.07

**Table 6 molecules-27-04430-t006:** The weight variation of Swiss albino female mice during the 14 days of free Mopp and MoP complex and control oral administration. The data are expressed as mean ± SD.

Concentration of Mopp and MoP	Weight (g) Day 1	Weight (g) Day 7	Weight (g) Day 14
50 mg/kg Polyphenols	21.60 ± 1.21	25.50 ± 0.32	27.0 ± 2.71
300 mg/kg Polyphenol Group 1	24.12 ± 0.10	28.15 ± 0.37	29.20 ± 1.33
300 mg/kg Polyphenol Group 2	23.65 ± 0.22	26.60 ±0.49	27.61 ± 1.42
2000 mg/kg Polyphenols	23.39 ± 0.28	26.1 ± 1.68	27.67 ± 2.79
50 mg/kg Phytosome Complex	24.67 ± 1.36	28.20 ± 1.32	28.50 ± 2.45
300 mg/kg Phytosome Complex	25.0 ± 1.31	27.67 ± 1.53	28.28 ± 0.27
300 mg/kg Phytosome Complex	22.0 ± 0.62	25.14 ± 1.02	25.67 ± 1.38
2000 mg/kg Phytosome Complex	26.65 ± 2.29	26.51 ± 1.92	27.06 ± 0.03
Control	23.00 ± 3.02	25.15 ± 2.58	27.11 ± 1.26

## Data Availability

All the data obtained are contained within the article.

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
