# Peer review of "Formulation, Optimization, and Evaluation of Moringa oleifera Leaf Polyphenol-Loaded Phytosome Delivery System against Breast Cancer Cell Lines"

_molecules, 2022, doi:10.3390/molecules27144430_

Round 1

Reviewer 1 Report

The manuscript provides an timely focus work for development and Evaluation of Moringa oleifera Leaf Polyphenols-Loaded Phytosome Delivery System against Breast Cancer Cell Lines. The work is well planned, However, due to signifying the purpose of research to improve the bioavailability through phytosomes encapsulation to enhance activity on 4T1 cancer cell lines won’t be suitable to attract high attention with the readers of the MDPI-Molecules because of following reasons:

Bioavailability needs to demonstrate in animal models which can be proved through pharmacokinetic studies.

And in-vitro drug dissolution studies are also required to correlate the significance of work.

As such, I recommend acceptance after considering the suggestions as described above.

Author Response

Comment 1: Bioavailability needs to demonstrate in animal models which can be proved through pharmacokinetic studies.

Response: We appreciate the response on this topic however; we only presented the work that was within the scope of our objectives

Comment 2: And in-vitro drug dissolution studies are also required to correlate the significance of work.

Response: This wasn’t within the scope of ours objectives; Our main objectives was to isolate Moringa oleifera polyphenols, formulate Moringa oleifera phytosomes and carryout in vitro antiproliferative activity of 4TI  Cancer cell lines. Another objective was to carry out in vivo safety of Moringa oleifera phytosomes.

Reviewer 2 Report

I have completed my evaluation of the manuscript by Jecinta Wanjiru et al. entitled “Formulation, Optimization, and Evaluation of Moringa oleifera Leaf Polyphenols-Loaded Phytosome Delivery System against Breast Cancer Cell Lines”.

Breast cancer is increasingly common around the world. This study showed that Moringa oleifera polyphenols-loaded phytosomes (Mop) prepared by nanoprecipitation method not only improved the bioavailability of leaf polyphenols (Mopp), but also had an inhibitory effect on the proliferation of 4T1 cancer cell lines. Thus, current study represents significant progress in breast cancer therapy. However, there are some limitations that prompt me suggest major revision to improve this manuscript!

Abstract

1.     Please read "MDPI Author Layout Style Guide" carefully. The abstract contains a summary of the entire paper and can be up to 200 words long with only one paragraph. https://www.mdpi.com/authors/layout

2.     Please define the abbreviations when they first appear in the manuscript!

3.     It is recommended to add "breast cancer" to the keywords.

4.     A Graphical Abstract should be added to show the aim of this research.

Introduction

1.     Please limit the Introduction to 4 paragraphs, only.

Results and Discussion

1.       It is recommended that section 2.1 and 2.3 be combined. The description of standard curves in Section 2.1 can be placed in the corresponding material methods section or in supplementary materials.

2.       Please replace Figure 2 and 3 with clearer pictures.

3.       Please unify the group names in the figure and table in the full text.

4.       Significance analysis of data is necessary.

5.       Line 164-166: There is a writing error here, please correct it.

6.       "Mo p" in Figure 6 is incorrectly written.

7.       There are many problems in Section 2.7. Please modify the group names in the chart to ensure the same context. Modify the color of the bar chart at "3.039" in figure 7, and check whether the information and units represented by the Y-axis in the chart are correct.

8.       Line 242-243: “while the selectivity index that measures the drug's ability to control cancer growth was 2.07±1.30, 12.79±0.50 and 5.34±0.13 μg/ml respectively as seen in Figure 7.”

Line 251-257: “Generally, the M.oleifera phytosomes inhibited the proliferation of 4T1 cancer cells. These findings thus indicated that a low concentration of M.oleifera loaded -polyphenols phytosomes at 0.14 μg/ml did not affect cell growth much, whereas the treatment with 0.41, 1.23, 3.70, 11.11, 33.33, 100 μg/ml of M.oleifera phytosomes inhibited the growth of 4T1 cells in a dose-dependent manner. There was a significant difference between the cytotoxicity of Mopp and MoP (p ≤ 0.05), and their activity was not comparable to that of the positive control.”

Please add the figure or table corresponding to the above data in this section. If supplementary materials have been provided, please mark the corresponding position.

9.       Line 250: “Doxorubicin was toxic to 4T1 cell 250 lines as the selectivity index was ≤ 3”, the description of this part is not precise.

10.   Line 260: Please change 39.94 to 39.84.

11.   Please modify the results and discussion sections, modify the structure order, clarify the logical relationship of the discussion sections, and avoid repetitive descriptions.

Materials and Methods

1.     Line 311/316: “with some modifications”/ ”with slight modifications”.

Is the method described here an improved method?

2.     Please supplement the experimental methods in section 3.10. What is the method of drug-loading residual content? If the content is too much, you can write in the supplementary materials.

3.     Please shorten the description in section 3.1. In this section, please add the Institutional Review Board Statement and approval number and approved date for studies involving humans or animals (for more details, see https://www.mdpi.com/ethics). In animal experiments, statistics generally require at least 6 data available for each group to be meaningful. There are usually no less than 10 mice in each group. In this study, there were only 3 mice in each group. Are the results of this experiment reliable?

Conclusions

1.     Please write the conclusion and argument directly. There is no need to elaborate on the background again. You can elaborate in the introduction or discussion.

      Although the wording along the manuscript is generally correct, I have found some passages that are difficult to understand. These minor problems could be easily solved if the authors have a colleague who is fluent in English and can proofread the writing.

Author Response

Abstract

Comment 1: Please read "MDPI Author Layout Style Guide" carefully. The abstract contains a summary of the entire paper and can be up to 200 words long with only one paragraph. https://www.mdpi.com/authors/layout

Response line 31 to 46:  Show summarised abstract of 200 words

Comment 2: Please define the abbreviations when they first appear in the manuscript!

Response line 35 to 36: shows added (3-[4,5-dimethylthiazol-2-yl]-2,5-diphenyltetrazole) MTT assay abbreviation when it first appear

Comment 3: It is recommended to add "breast cancer" to the keywords.

Response line 47 keywords now reads: polyphenols; natural nanoparticles; Moringa oleifera; antiproliferation; breast cancer

Comment 4: A Graphical Abstract should be added to show the aim of this research.

Response line 48: Show added graphical abstract

Introduction

       Comment 1:  Please limit the Introduction to 4 paragraphs, only.

       Response in introduction line 50 to 102: Summarized to 4 paragraph

Results and Discussion

        Comment 1: It is recommended that section 2.1 and 2.3 be combined. The description of standard curves in Section 2.1 can be placed in the corresponding material methods section or in supplementary materials.

        Response (a) line 108 to 163: Shows combined section 2.1 to 2.3

       Response (b) line 376 to 385: Shows the description of standard curve that have been placed in corresponding material methods section

Comment 2: Please replace Figure 2 and 3 with clearer pictures.

        Response line 179 and line 186: Show replaced with now called Figure 3 and 4 with clearer pictures

       Comment 3:  Please unify the group names in the figure and table in the full text.

       Response:  I have unified the group names in Figure 1 to 7 and Table 1 to 6

       Comment 4: Significance analysis of data is necessary.

       Response in line 306 to 310: Significance of data added

       Comment 5 Line 164-166: There is a writing error here, please correct it.

Response in now line 161 to 163: Shows corrected error and now read “Previous studies have reported that the formulations % EE depends on drug solubility and bonding interaction leading to matrix formation. Therefore, the present data support results from the earlier studies by Pal et al. [36], in which a higher % EE for polyphenols is demonstrated.”

       Comment 6: "Mo p" in Figure 6 is incorrectly written.

       Response line 242: "Mo p" in Figure 6 is correctly written as "Mop"

       Comment 7: There are many problems in Section 2.7. Please modify the group names in the chart to ensure the same context. Modify the color of the bar chart at "3.039" in figure 7, and check whether the information and units represented by the Y-axis in the chart are correct.

       Response (a) in section 2.7: The group name have been modified to ensure same context

       Response (b) in section 2.7 in line 285: The colour of bar chart in Figure 7 now modified and decimal place replaced with 3.04 from 3.039

       Response (c) in section 2.7 in line 285: The units on Y-axis in now reads IC50 of the extracts in µg/mL

       Comment 8 (a): Line 242-243: “while the selectivity index that measures the drug's ability to control cancer growth was 2.07±1.30, 12.79±0.50 and 5.34±0.13 μg/mL respectively as seen in Figure 7.”

       Response in now line 283 to 284: Corrected and now reads ‘The selectivity index for doxorubicin, MoP and Mopp was 2.07±1.30, 12.79±0.50 and 5.34±0.13 μg/mL respectively as seen in Figure 7’

       Comments 8 (b): Line 251-257: “Generally, the Moringa oleifera phytosomes inhibited the proliferation of 4T1 cancer cells. These findings thus indicated that a low concentration of Moringa oleifera loaded-polyphenols phytosomes at 0.14 μg/mL did not affect cell growth much, whereas the treatment with 0.41, 1.23, 3.70, 11.11, 33.33, 100 μg/mL of Moringa oleifera phytosomes inhibited the growth of 4T1 cells in a dose-dependent manner. There was a significant difference between the cytotoxicity of Mopp and MoP (p ≤ 0.05), and their activity was not comparable to that of the positive control. Please add the figure or table corresponding to the above data in this section. If supplementary materials have been provided, please mark the corresponding position.

       Response line 289: Figure 8 that corresponds to 100, 33.33, 11.11, 3.70, 1.23, 0.41 and 0.14 μg/mL concentrations added.

       Comment 9:  Line 250: “Doxorubicin was toxic to 4T1 cell 250 lines as the selectivity index was ≤ 3”, the description of this part is not precise.

       Response in now line 294 to 296: The description is now precise and now reads “Doxorubicin was toxic to 4T1 cell lines as the selectivity index was ≤ 3 and this is because according National cancer institute, selectivity index below 3 indicates indicates higher cellular toxicity.” 

      Comment 10: Line 260 Please change 39.94 to 39.84.

       Response in now line 283:  Corrected and now reads 39.84±0.10

      Comment 11: Please modify the results and discussion sections, modify the structure order, clarify the logical relationship of the discussion sections, and avoid repetitive descriptions.

      Response line 108 to 163:  Structure modified and repetitive descriptions removed

      Materials and Methods

    Comment 1: Line 311/316: “with some modifications”/ ”with slight modifications”. Is the method described here an improved method

    Response: Yes, this are improved method

     Comment 2:   Please supplement the experimental methods in section 3.10. What is the method of drug-loading residual content? If the content is too much, you can write in the supplementary materials.

     Response line 412: drug loading residual content method added

      Comment 3:  Please shorten the description in section 3.1. In this section, please add the Institutional Review Board Statement and approval number and approved date for studies involving humans or animals (for more details, see https://www.mdpi.com/ethics). In animal experiments, statistics generally require at least 6 data available for each group to be meaningful. There are usually no less than 10 mice in each group. In this study, there were only 3 mice in each group. Are the results of this experiment reliable?

      Response (a) section 3.1 line 335 to 340: Description section summarised

      Response (b) section 3.1 in line 342 to 346: Section 3.2 added describing institutional review board statement and approval numbers date for this study

      Response (c) section 3.1: This was mistakenly written, we used 7 mice in each group but we have corrected this

Conclusions

      Comment 1:  Please write the conclusion and argument directly. There is no need to elaborate on the background again. You can elaborate in the introduction or discussion.

      Response in line 464 to 474: Conclusion summarized and written directly

      Comment 2: Although the wording along the manuscript is generally correct, I have found some passages that are difficult to understand. These minor problems could be easily solved if the authors have a colleague who is fluent in English and can proofread the writing.

      Response: The work was Proofread by a fluent colleague

Reviewer 3 Report

The authors presented an interesting type of phytosomes loaded with Moringa oleifera polyphenols. The study is very interesting, but presents as a draft of article.

First of all, the authors must build a logic chain of research. Each stage of the study as well the purpose of the study should be explained and justified. In this way the authors need to improve the manuscript.

There are a number of suggestions which will improve the presentation of results:

1) Figure 1 shows absorbance at 425 nm but the description is “… by spectrophotometry at 760 nm.”(line 110).

2) When plotting the standard curve, optical density (absorbance) values not higher than 1 are used. So the authors should remeasure and recalculate the standard curve in such condition, where the highest concentration of standard gallic acid corresponds to optical density with value 1. And other points on the curve (at least 7-10) have values of optical density in the range of 0.05-1. This will provide an accurate standard calibration curve.

3) DLS technique allows to measure not the particle size, but the size of the hydrodynamic diameter of a particle in a solution. Figures 2 and 3 can be presented in a Supporting information, the obtained values themselves (the hydrodynamic diameter and zeta-potential) should be presented and described in the text. The data in Figure 2 is not presented correctly. It is necessary to make from 3 to 5 measurements of the hydrodynamic diameter and present the final data as mean ± standard deviation in nanometers (accurate to a nanometer, without tenths or hundredths) where mean is an average value of Z-Average in this series of measurements.

4) According to the Figure 3 the correct value of zeta-potential is -40.1±7.3 mV. Please correct obtained data.

5) Table 3: please leave only two decimal places after the dot. In the “Cumulative Percentage Drug Release…” column the SD is missing.

6) Table 5: units should be added.

7) The "Discussion" section should be presented in more detail and expanded. 

Author Response

Comment 1: Figure 1 shows absorbance at 425 nm but the description is “… by spectrophotometry at 760 nm.”(line 110).

Response in now line 130: Absorbance error corrected in Figure 1 and now reads absorbance 760 nm

Comment 2: When plotting the standard curve, optical density (absorbance) values not higher than 1 are used. So the authors should remeasure and recalculate the standard curve in such condition, where the highest concentration of standard gallic acid corresponds to optical density with value 1. And other points on the curve (at least 7-10) have values of optical density in the range of 0.05-1. This will provide an accurate standard calibration curve.

Response: The concentration of standard gallic acid and the optical density followed was in accordance to a study entitled,Bioactive Compounds in Salicornia patula Duval-Jouve: A Mediterranean Edible Euhalophyte” by Sánchez-Gavilán et al., 2021 that reported the optical density above 1 in Figure 1.

Comment 3: DLS technique allows to measure not the particle size, but the size of the hydrodynamic diameter of a particle in a solution. Figures 2 and 3 can be presented in a Supporting information, the obtained values themselves (the hydrodynamic diameter and zeta-potential) should be presented and described in the text. The data in Figure 2 is not presented correctly. It is necessary to make from 3 to 5 measurements of the hydrodynamic diameter and present the final data as mean ± standard deviation in nanometers (accurate to a nanometer, without tenths or hundredths) where mean is an average value of Z-Average in this series of measurements.

Response 166 to 178: Dynamic light scattering measurement explained clearly

Comment 4: According to the Figure 3 the correct value of zeta-potential is -40.1±7.3 mV. Please correct obtained data.

Response in now line 169: Obtained value corrected to read a zeta potential value of -40.1±7.3 mV

Comment 5: Table 3: please leave only two decimal places after the dot. In the “Cumulative Percentage Drug Release…” column the SD is missing.   

Response (a) Table 3 in line 226: Table 3 decimal place corrected to 2

Response (b) Table 3 in line 226: Standard deviation added

Comment 6: Table 5: units should be added.

Response line 263: Table 5 Units added and now reads Average Particle Size in nm and Zeta Potential in mV

Comment 7: The "Discussion" section should be presented in more detail and expanded. 

Response to discussion: Discussion part now presented in more detail for example in line 146 to 154, 166 to 178 and the other part of discussion.

Round 2

Reviewer 1 Report

Manuscript cannot be accepted saying the experiment suggested was not within the scope of objective.

At least in-vitro dissolution and release studies can be performed which is one of the basic experiment for formulations and can be finished in one day.

Author Response

Comment: At least in-vitro dissolution and release studies can be performed which is one of the basic experiment for formulations and can be finished in one day.

Response (a) release profile line 216-257: Thank you for this comment, the dissolution studies had been captured as release profile studies, the studies had been done and included in the original manuscript line 216-257 and it reads:

 “ 2.4 In vitro drug release. The in vitro drug release of optimized MoP complex and Mopp is shown in Figure 6 while the drug release for different kinetic models with their correlation coefficient was shown in Tables 3 and 4.”

Also Line 257-273 added and now reads:

In our study in vitro release testing” was used as a measure of drug dissolution as documented previously by Shen and Burgess [26]. This is considered an important tool for quality control purposes as well as for prediction of the in vivo performance of drug delivery involving nanocarrier systems and has become an essential quality control test of drug development since it was officially adopted in the United State Pharmacopeia (USP) in 1970 [27-28]. There is no standard pharmacopeial/regulatory in vitro dissolution/release test currently available for nanoparticulate systems [26]. However, extensive efforts have been made to develop suitable in vitro dissolution/release testing methods for nanoparticulate delivery systems. Current methods are broadly divided into three categories namely; the membrane diffusion methods (such as the dialysis methods), sample and separation methods, and the continuous flow methods.

Membrane diffusion methods (dialysis methods) are the most widely investigated for the in vitro dissolution release testing of the nanoparticulate systems. In these methods, the nanoparticulate systems are separated from the release medium through dialysis membranes that are permeable to the free drug but impermeable to the nanoparticles. Dialysis methods have been widely used to investigate in vitro drug dissolution/release profiles of liposomes [29-30], emulsions  [31], polymeric nanoparticles  [32-33], as well as lipid nanocarriers  [34].”

Response (b) bioaccessibility and bioavailability line 437-471 and 274-286:  Thank you very much for this comment. We apologize for this omission in the manuscript. In our studies we had done in vitro bioaccessibility but not in vivo bioavailability.

In vitro bioavailability has been added under the heading in vitro  bioaccessibility

Results, Line 274-286 now reads:

2.5 In vitro bioaccessibility. As shown in Figure 7, there was a significant difference in percentage bioaccessibility of polyphenols in the GIT when TPC of Mopp was compared to MoP (p value 0.001) after in vitro digestion. Therefore, these results proved the successful effect of phytosomes as a carrier (Fig. 7).

Figure 7: Bioaccessibility comparison of Mopp and MoP after exposure to simulated gastro intestinal conditions

The free Mopp bioaccesibility mean percentage was 26.95±0.02, showing extensive degradation whereas the MoP bioaccesibility mean percent was 66.98±0.01%; indicating a better stability than the Mopp. The low bioaccesibility of Mopp was due to compromised conditions (digestive enzymes and the acidic environment) of the gastro intestinal tract. These findings are in line with previous report where low bioaccesibility of free phenolic compounds were seen after in vitro digestion as compared to encapsulated polyphenols [35,36].This might be attributed to low absorption due to larger molecular weight and thus limited bioavailability.”

Methodology, Line 437- 471 now reads:

3.11 In vitro biaccessibility determination of MoP and Mopp. The MoP and Mopp were evaluated for their bioaccessibility under simulated gastro intestinal (GIT) model consisting the mouth, stomach and intestine according to Grgić et al. [36] method.  The prepared samples were then exposed to the simulated gastric and small intestine phases.

3.11.1 Simulated salivary fluid in Mouth Phase

Simulated salivary fluid  phase (SSF) was prepared using  0.328 g/L ammonium nitrate,1.594 g/L sodium chloride, 0.202 g/L potassium chloride, potassium phosphate 0.636 g/L, urea 0.198 g/L, 0.308 g/Lpotassium citrate,  0.146 g/L lactic acid sodium salt and 5 g/L Porcine gastric Mucin Type II. An aliquot of  4mL of each extract was mixed with 4 mL of simulated saliva and the pH of the mixture adjusted to 6.8. The mixture was shaken continuously for 10 minutes at 100 rev while maintaining at   37℃.

3.11.2 Simulated gastric fluid (SGF)

The SGF was prepared with a slight modification of Shah et al. [35]; Grgić et al. [36] method. Two grams of sodium chloride was dissolved in 10 mL of 7.0 mL of hydrochrolic acid (420 g/L). Pepsin (3.2g) were dissolved in 1L of double distilled water and the pH adjusted to 1.2 using 1MHCl. The sample from mouth phase mimicking the bolus was mixed with SGF phase at a ratio of 50:50. The pH of the 2 phases mixture was adjusted to 2.0 using 1M NaOH and incubated at 37°C for 2 h with continuous shaking, at speed of 100 rev/min.

3.11.3 Small intestinal phase

About 15ml of the digested sample from the gastric phase was mixed with 8.25mL simulated intestinal buffer solution.  An aliquot of 1.87 mL fresh bile extract, 30 µL of 0.3M calcium chloride and 3.75 mL of pancreatin solution were also added and the volume topped up to 30ml using deionized water. The temperature was maintained at 37°Cand pH adjusted to 7.0 with 1M NaOH. Approximately 1.5 mL lipase suspension (at concentration of 60 mg/mL) was dissolved in phosphate buffer saline (PBS) and added to the mixture. The mixture was then allowed to shake for 2 h while monitoring the pH (pH was maintained at 7.0) to mimic intestinal digestion process. 0.25M NaOH was used to neutralize the fatty acids released from the lipid digestion while maintaining the pH of 7.0.  The mixture was then shaken   at 100 rpm, for 6 h at 37°C.

3.11.4 Measurement of bioaccessibility

At the end of in vitro digestion, digested sample was used to measure the percentage bioaccessibility. To the MoP sample, 1% Triton X-100 was added to rupture the lipid membrane, and the mixture vortexed and later centrifuged at 20,000 rpm, for 30 min at 4ËšC. The supernatant was collected and filtered. The filtrate was then fractioned and phenolic compounds solubilized. The total phenolic content was quantified and bioaccessibility calculated as follows;

???????????????? (%) = ??????????∗ 100.”

Reviewer 2 Report

The revised manuscript has been highly improved based on the reviewers comments. It can be accepted in Molecules. 

Author Response

Comments and Suggestions for Authors: The revised manuscript has been highly improved based on the reviewers’ comments. It can be accepted in Molecules.

Response: We greatly appreciate the reviewer’s previous comments and suggestions that helped improved the manuscript. Your positive consideration for publication is also appreciated.

Reviewer 3 Report

The article can be accepted after minor corrections. 

Please round up to 1-2 significant figures (depending on instrument error) all presented results (DLS data, mass measurements, etc.) 

Author Response

Comment 1: Please round up to 1-2 significant figures (depending on instrument error) all presented results (DLS data, mass measurements, etc.) 

Response on DLS significant figures in line 177, 184, 331 and 332,: Significant figures rounded up to 2